# River plastic emissions to the world's oceans

Laurent C.M. Lebreton[1,2], Joost van der Zwet[3], Jan-Willem Damsteeg[1], Boyan Slat[1], Anthony Andrady[4] & Julia Reisser[1]

Plastics in the marine environment have become a major concern because of their persistence at sea, and adverse consequences to marine life and potentially human health. Implementing mitigation strategies requires an understanding and quantification of marine plastic sources, taking spatial and temporal variability into account. Here we present a global model of plastic inputs from rivers into oceans based on waste management, population density and hydrological information. Our model is calibrated against measurements available in the literature. We estimate that between 1.15 and 2.41 million tonnes of plastic waste currently enters the ocean every year from rivers, with over 74% of emissions occurring between May and October. The top 20 polluting rivers, mostly located in Asia, account for 67% of the global total. The findings of this study provide baseline data for ocean plastic mass balance exercises, and assist in prioritizing future plastic debris monitoring and mitigation strategies.

[1] The Ocean Cleanup Foundation, Martinus Nijhofflaan 2, Delft 2624 ES, The Netherlands. [2] The Modelling House, 66b Upper Wainui Road, Raglan 3297, New Zealand. [3] HKV Consultants, PO Box 2120, Lelystad 8203 AC, The Netherlands. [4] Department of Chemical and Biomolecular Engineering, North Carolina State University, Campus Box 7905, Raleigh, North Carolina 27695, USA. Correspondence and requests for materials should be addressed to L.C.M.L. (email: laurent.lebreton@theoceancleanup.com).

Plastics are increasingly used worldwide in a wide variety of applications with global production exceeding 300 million tonnes per year since 2014 (ref. 1). Because of their durability, low-recycling rates, poor waste management and maritime use, a significant portion of the plastics produced worldwide enters and persists in marine ecosystems[2]. This includes shoreline, seabed, water column and sea surface environments of the world's oceans[3]. The release of plastics into the marine environment occurs through a variety of pathways, including river and atmospheric transport, beach littering and directly at sea via aquaculture, shipping and fishing activities[4]. A comprehensive risk assessment of this relatively new type of marine contamination requires defining, understanding and quantifying emissions both geographically and temporally. This knowledge helps in refining our understanding of marine plastic pollution sources and pathways, while working towards an estimated global budget for ocean plastics. It also assists in identifying the critical locations and seasons of plastic releases, supporting the implementation of cost-effective monitoring and source mitigation efforts.

Land-based sources, as opposed to marine-based sources, are considered the dominant input of plastics into oceans[4]. While a quantification of land-based inputs from coastal populations (<50 km from coastline) worldwide already exists[5], no global assessment of contributions from inland populations through riverine systems has been proposed to date. Only a few local studies have reported levels of plastic contamination in freshwater systems worldwide[6]. Such freshwater studies generally focus on micro-plastics (< 5 mm length) contaminating sediment and waters of lakes and rivers. Sampling design and devices, as well as reported units (for example, pieces per m$^2$, grams per m$^3$) vary substantially between different assessments[7]. Overall, observed plastic concentrations differ by several orders of magnitude in between sampled rivers[7], with studies suggesting that population density[8,9], levels of urbanization and industrialization within catchment areas[10,11], rainfall rates[10,12] and the presence of artificial barriers such as weirs and dams[13] play a significant role in resulting rates of river-based plastic inputs into the ocean.

A study in California[12] was the first to report levels of micro-plastics in river surface waters, with sampling in Los Angeles River, San Gabriel River and tributary Coyote Creek. The report found substantial temporal variations in plastic contamination

levels. For a given location, the study found up to three orders of magnitude differences between plastic concentrations measured at different time periods. These variations were mostly explained by events of dry and wet weather, implying that runoff plays an important role in the transport of plastics into freshwater systems. In recent years, more studies sampled plastic in surface waters of rivers. In Europe, studies estimated that the Danube River releases 530–1,500 tonnes of plastic into the Black Sea annually[8,9]. Another European study estimated that 20–31 tonnes flows into the North Sea every year from the Rhine River[8], with different locations along this river demonstrating the presence of significant sources (for example, wastewater treatment plants, tributaries) and sinks (for example, weirs; ref. 13). In the Italian Po River, sampled concentrations differed by one order of magnitude between winter and spring[14], emphasising seasonality of freshwater contamination in rivers.

Surface plastic concentrations showed a statistically significant correlation with human population densities and proportion of urban development in catchments around the Chesapeake Bay (USA)[10]. The study also reported the highest measured micro-plastic concentrations (up to 1.6 milligrams per cubic metre for 1.73 particles per cubic metre) occurring after major rain events at three of the four monitoring sites. In South America, monitoring from scientists and volunteers at the Elqui, Maipo, Biobio and Maule Rivers, demonstrated a consistent pattern between numerical concentration of micro-plastics (range 0.05–0.74 particles per cubic metre) and the presence of litter deposition on riversides[15]. In Asia, sampling of micro-plastics on beaches suggested that the nearby Pearl River is a major source of plastic pollution for the region[16]. Surface samplings at the Chinese Yangtze River mouth showed considerably higher plastic concentrations than any other sampled river worldwide[17] with a reported 4,137 particles per cubic metre. The significant differences between sampled estuarine concentrations and nearshore monitoring in the area confirmed that the Yangtze River is a major regional source of plastic input into the marine environment.

Here we provide a global estimate of river plastic inputs into the world's oceans, considering both the seasonality and spatial variability of local sources. Our global model uses geospatial data of population density[18,19], rates of mismanaged plastic waste (MPW) production per inhabitant and per country[5,20], monthly

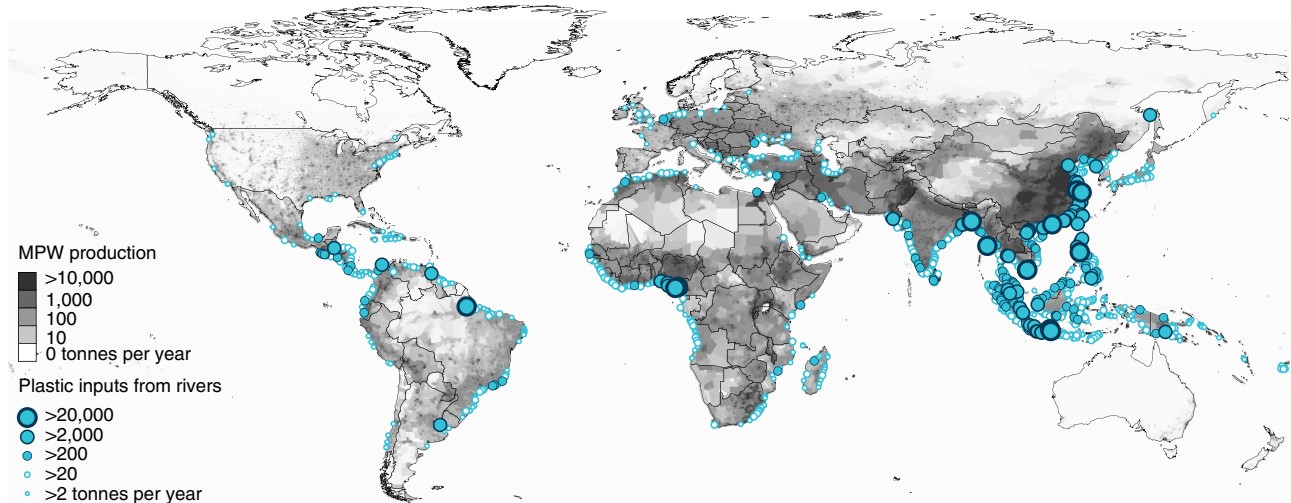

**Figure 1 | Mass of river plastic flowing into oceans in tonnes per year.** River contributions are derived from individual watershed characteristics such as population density (in inhab km$^{-2}$), mismanaged plastic waste (MPW) production per country (in kg inhab$^{-1}$ d$^{-1}$) and monthly averaged runoff (in mm d$^{-1}$). The model is calibrated against river plastic concentration measurements from Europe, Asia, North and South America.

**Table 1 | Top 20 polluting rivers as predicted by the global river plastic inputs model.**

| Catchment | Country | Lower mass input estimate (t yr$^{-1}$) | Midpoint mass input estimate (t yr$^{-1}$) | Upper mass input estimate (t yr$^{-1}$) | Total catchment surface area (km$^2$)[21] | Yearly average discharge (m$^3$ s$^{-1}$)[21] |
|---|---|---|---|---|---|---|
| Yangtze | China | $3.10 \times 10^5$ | $3.33 \times 10^5$ | $4.80 \times 10^5$ | $1.91 \times 10^6$ | $1.58 \times 10^4$ |
| Ganges | India, Bangladesh | $1.05 \times 10^5$ | $1.15 \times 10^5$ | $1.72 \times 10^5$ | $1.57 \times 10^6$ | $2.08 \times 10^4$ |
| Xi | China | $6.46 \times 10^4$ | $7.39 \times 10^4$ | $1.14 \times 10^5$ | $3.89 \times 10^5$ | $5.53 \times 10^3$ |
| Huangpu | China | $3.35 \times 10^4$ | $4.08 \times 10^4$ | $6.73 \times 10^4$ | $2.62 \times 10^4$ | $4.04 \times 10^2$ |
| Cross | Nigeria, Cameroon | $3.38 \times 10^4$ | $4.03 \times 10^4$ | $6.5 \times 10^4$ | $2.38 \times 10^4$ | $2.40 \times 10^2$ |
| Brantas | Indonesia | $3.23 \times 10^4$ | $3.89 \times 10^4$ | $6.37 \times 10^4$ | $1.11 \times 10^4$ | $8.18 \times 10^2$ |
| Amazon | Brazil, Peru, Columbia, Ecuador | $3.22 \times 10^4$ | $3.89 \times 10^4$ | $6.38 \times 10^4$ | $5.91 \times 10^6$ | $1.40 \times 10^5$ |
| Pasig | Philippines | $3.21 \times 10^4$ | $3.88 \times 10^4$ | $6.37 \times 10^4$ | $4.07 \times 10^3$ | $2.07 \times 10^2$ |
| Irrawaddy | Myanmar | $2.97 \times 10^4$ | $3.53 \times 10^4$ | $5.69 \times 10^4$ | $3.77 \times 10^5$ | $5.49 \times 10^3$ |
| Solo | Indonesia | $2.65 \times 10^4$ | $3.25 \times 10^4$ | $5.41 \times 10^4$ | $1.58 \times 10^4$ | $7.46 \times 10^2$ |
| Mekong | Thailand, Cambodia, Laos, China, Myanmar, Vietnam | $1.88 \times 10^4$ | $2.28 \times 10^4$ | $3.76 \times 10^4$ | $7.74 \times 10^5$ | $6.01 \times 10^3$ |
| Imo | Nigeria | $1.75 \times 10^4$ | $2.15 \times 10^4$ | $3.61 \times 10^4$ | $7.92 \times 10^3$ | $2.79 \times 10^2$ |
| Dong | China | $1.57 \times 10^4$ | $1.91 \times 10^4$ | $3.17 \times 10^4$ | $3.33 \times 10^4$ | $8.54 \times 10^2$ |
| Serayu | Indonesia | $1.33 \times 10^4$ | $1.71 \times 10^4$ | $2.99 \times 10^4$ | $3.71 \times 10^3$ | $3.70 \times 10^2$ |
| Magdalena | Colombia | $1.29 \times 10^4$ | $1.67 \times 10^4$ | $2.95 \times 10^4$ | $2.61 \times 10^5$ | $5.93 \times 10^3$ |
| Tamsui | Taiwan | $1.16 \times 10^4$ | $1.47 \times 10^4$ | $2.54 \times 10^4$ | $2.68 \times 10^3$ | $1.08 \times 10^2$ |
| Zhujiang | China | $1.09 \times 10^4$ | $1.36 \times 10^4$ | $2.31 \times 10^4$ | $4.01 \times 10^3$ | $1.33 \times 10^2$ |
| Hanjiang | China | $1.03 \times 10^4$ | $1.29 \times 10^4$ | $2.19 \times 10^4$ | $2.95 \times 10^4$ | $7.35 \times 10^2$ |
| Progo | Indonesia | $9.80 \times 10^4$ | $1.28 \times 10^4$ | $2.29 \times 10^4$ | $2.24 \times 10^3$ | $2.79 \times 10^2$ |
| Kwa Ibo | Nigeria | $9.29 \times 10^4$ | $1.19 \times 10^4$ | $2.08 \times 10^4$ | $3.63 \times 10^3$ | $1.92 \times 10^2$ |

Input rate estimates (in t yr$^{-1}$) are representative of mismanaged plastic waste (MPW) production and catchment runoff. A lower, midpoint and upper estimate is calculated based on three regression analyses accounting for uncertainties in our field observations data set.

catchment runoff[21,22], as well as the presence of artificial barriers (for example, dams and weirs) that act as particle sinks[23,24]. The correlation found between model outputs and field observations suggests that the model presented here is able to describe a significant proportion of the spatio-temporal variations on levels of plastic contamination in freshwater systems worldwide. We predict a global annual input of plastic from rivers into the oceans ranging from 1.15 to 2.41 million tonnes with a dominant contribution from rivers of the Asian continent.

## Results

**River plastic mass inputs to oceans.** We estimated that between 1.15 and 2.41 million tonnes of plastic currently flows from the global riverine system into the oceans every year (Fig. 1). The top 20 polluting rivers were mostly located in Asia (Table 1) and accounted for more than two thirds (67%) of the global annual input while covering 2.2% of the continental surface area and representing 21% of the global population. Furthermore, the top 122 polluting rivers (4% of total landmass surface area and 36% of global population) contributed for >90% of the plastic inputs with 103 rivers located in Asia, eight in Africa, eight in South and Central America, and one in Europe.

Our model was calibrated against reported micro- and macro-plastic concentrations from river surface waters of Europe, Asia, North and South America (Table 2). The studies used for this assessment report only a fraction of the wide plastic debris size spectrum: from particles larger than their net mesh sizes (typically 0.3 mm) to objects smaller than the aperture size of their sampling devices (typically 0.5 m). Therefore, our global river plastic input estimate is conservative, as we neglect the contribution of debris outside the sampled size range and below the surface sampling devices. Both buoyant and non-buoyant river plastics can be suspended throughout the water column and transported towards the sea due to turbulent river flows and large flood events[25].

River plastic mass concentrations from the literature (Table 3) demonstrated a statistically significant, positive correlation with the product of annual MPW production rates inside a catchment area downstream of artificial barriers and monthly averaged catchment runoff (Pearson product-moment test, $r = 0.4132$, $P < 0.05$, $n = 29$). The correlation test was conducted after removing measurement data from the Yangtze River, considered as an outlier. When including data from the Yangtze River, we logically obtained a much higher, yet biased correlation ($r = 0.99$, $P < 0.05$, $n = 30$).

The product of annual MPW production inside a catchment area downstream of artificial barriers and monthly averaged catchment runoff for the month corresponding to the sampling period was used to formulate an empirical model. Uncertainties related to estimating daily flux inputs from concentration measurements were considered, while assessing a linear regression. We determined three parametric equations corresponding to lower, midpoint and upper model estimates to best predict estimation ranges from observations and account for these uncertainties (Supplementary Table 1). Our midpoint calculations demonstrated a good relationship with mass flux inputs derived from measurements by orders of magnitude with a coefficient of determination $r^2 = 0.93$ ($n = 30$; Fig. 2).

Estimated plastic releases from Asian rivers represented 86% of the total global input. A considerably high-population density combined with relatively large MPW production rates and episodes of heavy rainfalls, resulted in this dominant contribution from the Asian continent, with an estimated annual input of 1.21 (range 1.00–2.06) million tonnes per year. Our model predicted that the Chinese Yangtze River is the largest contributing catchment, with an annual input of 0.33 (range 0.31–0.48) million tonnes of plastic discharged into the East China Sea, followed by the Ganges River catchment, between India and Bangladesh, with a computed input of 0.12 (range 0.10–0.17) million tonnes per year. The combined input of the Xi, Dong and Zhujiang Rivers in China all flowing into the South China Sea at the Pearl River delta, was estimated at 0.106 (range 0.091–0.169)

**Table 2 | List of observational studies used in the calibration of the global river plastic inputs model.**

| References | River(s) | Region | No of records | Numerical Concentration | Mass Concentration | Sampling method |
|---|---|---|---|---|---|---|
| Lechner et al.[9] | Danube | Europe | 2 | Macro & micro | Macro & micro | Stationary conical driftnets (0.5 m diameter, 0.5 mm mesh). |
| Van der Wal et al.[8] | Danube, Rhine, Po | Europe | 4 | Macro & micro | Macro | Manta trawl net (0.33 mm mesh) and WFW sampler (3.2 mm mesh). |
| Vianello et al.[14] | Po | Europe | 2 | Micro | NA | Neuston net (0.33 mm mesh). |
| Dris et al.[44] | Seine | Europe | 1 | Micro | NA | Manta trawl (0.33 mm mesh). |
| Zhao et al.[17] | Yangtze | Asia | 1 | Micro | NA | Survey at estuary using a submerged pump (0.032 mm steel sieve). |
| Rech et al.[15] | Elqui, Maipo, Biobio, Maule | South America | 4 | Micro | NA | 27 cm by 10.5 cm neuston net (1 mm mesh). |
| Yonkos et al. [10] | Patapsco, Magothy, Rhode, Corsica | North America | 16 | Micro | Micro | Survey at estuary. 1-2 km surface trawl using 70 cm by 15 cm manta net (0.33 mm mesh). Numerical concentration reported by surface unit area. |

As shown here, not all studies ($n = 30$ records from 13 rivers) reported both numerical (for example, # m$^{-3}$) and mass (for example, g m$^{-3}$) concentrations for both macro- ($>5$ mm) and micro-plastics ($< 5$ mm). When numerical and/or mass concentrations are not reported, we extrapolated values from reported micro-plastic concentrations (see main manuscript and Table 3).

**Table 3 | Plastic concentration in sampled rivers and watershed characteristics.**

| River | Micro (# m$^{-3}$) | Macro (# m$^{-3}$) | Micro (g m$^{-3}$) | Macro (g m$^{-3}$) | Total (g m$^{-3}$) | $M_{mpw}$ (t yr$^{-1}$) | $R$ (mm d$^{-1}$) |
|---|---|---|---|---|---|---|---|
| Danube[9] | $8.23 \times 10^{-1}$ | $1.2 \times 10^{-1}$ | $9.8 \times 10^{-3}$ | $1.10 \times 10^{-3}$ | $1.09 \times 10^{-2}$ | $5.96 \times 10^{5}$ | $4.3 \times 10^{-1}$ |
| Danube[9] | $4.00 \times 10^{-2}$ | $1.5 \times 10^{-2}$ | $2.0 \times 10^{-3}$ | $2.00 \times 10^{-4}$ | $2.20 \times 10^{-3}$ | $5.96 \times 10^{5}$ | $4.3 \times 10^{-1}$ |
| Danube[8] | $1.06 \times 10^{1}$ | $3.7 \times 10^{-1}$ | $3.2 \times 10^{-2}$ | $3.80 \times 10^{-2}$ | $6.98 \times 10^{-2}$ | $5.96 \times 10^{5}$ | $3.4 \times 10^{-1}$ |
| Rhine[8] | $4.92 \times 10^{0}$ | $5.0 \times 10^{-2}$ | $1.5 \times 10^{-2}$ | $7.90 \times 10^{-2}$ | $9.38 \times 10^{-2}$ | $1.62 \times 10^{5}$ | $3.2 \times 10^{-1}$ |
| Rhine[8] | $1.85 \times 10^{0}$ | $4.2 \times 10^{-2}$ | $5.6 \times 10^{-3}$ | $7.70 \times 10^{-3}$ | $1.33 \times 10^{-2}$ | $1.62 \times 10^{5}$ | $2.8 \times 10^{-1}$ |
| Po[8] | $1.46 \times 10^{1}$ | $3.2 \times 10^{-2}$ | $4.4 \times 10^{-2}$ | $3.80 \times 10^{-2}$ | $4.76 \times 10^{-2}$ | $1.63 \times 10^{4}$ | $1.9 \times 10^{0}$ |
| Po[14] | $1.00 \times 10^{0}$ | $4.0 \times 10^{-2}$ | $3.0 \times 10^{-3}$ | $6.87 \times 10^{-3}$ | $9.87 \times 10^{-3}$ | $1.63 \times 10^{4}$ | $1.4 \times 10^{0}$ |
| Po[14] | $1.22 \times 10^{1}$ | $4.9 \times 10^{-1}$ | $3.7 \times 10^{-2}$ | $8.38 \times 10^{-2}$ | $1.20 \times 10^{-1}$ | $1.63 \times 10^{4}$ | $1.1 \times 10^{0}$ |
| Seine[44] | $3.75 \times 10^{-1}$ | $1.5 \times 10^{-2}$ | $1.1 \times 10^{-3}$ | $2.58 \times 10^{-3}$ | $3.70 \times 10^{-3}$ | $2.04 \times 10^{4}$ | $1.2 \times 10^{-1}$ |
| Elqui[15] | $1.29 \times 10^{-1}$ | $5.2 \times 10^{-3}$ | $3.9 \times 10^{-4}$ | $8.85 \times 10^{-4}$ | $1.27 \times 10^{-3}$ | $6.47 \times 10^{2}$ | $3.5 \times 10^{-2}$ |
| Maipo[15] | $6.47 \times 10^{-1}$ | $2.6 \times 10^{-2}$ | $1.9 \times 10^{-3}$ | $4.45 \times 10^{-3}$ | $6.39 \times 10^{-3}$ | $2.38 \times 10^{4}$ | $6.0 \times 10^{-2}$ |
| BioBio[15] | $5.00 \times 10^{-2}$ | $2.0 \times 10^{-2}$ | $1.5 \times 10^{-4}$ | $3.44 \times 10^{-4}$ | $4.94 \times 10^{-4}$ | $3.26 \times 10^{3}$ | $9.4 \times 10^{-2}$ |
| Maule[15] | $7.40 \times 10^{-1}$ | $3.0 \times 10^{-2}$ | $2.2 \times 10^{-3}$ | $5.09 \times 10^{-3}$ | $7.31 \times 10^{-3}$ | $3.12 \times 10^{3}$ | $1.2 \times 10^{-1}$ |
| Patapsco[10] | $8.72 \times 10^{0}$ | $3.5 \times 10^{-1}$ | $5.1 \times 10^{-4}$ | $5.99 \times 10^{-2}$ | $6.05 \times 10^{-2}$ | $7.80 \times 10^{2}$ | $2.4 \times 10^{-1}$ |
| Patapsco[10] | $1.99 \times 10^{0}$ | $7.9 \times 10^{-2}$ | $1.6 \times 10^{-3}$ | $1.36 \times 10^{-2}$ | $1.52 \times 10^{-2}$ | $7.80 \times 10^{2}$ | $2.9 \times 10^{-1}$ |
| Patapsco[10] | $3.99 \times 10^{-1}$ | $1.6 \times 10^{-2}$ | $7.1 \times 10^{-5}$ | $2.74 \times 10^{-3}$ | $2.81 \times 10^{-3}$ | $7.80 \times 10^{2}$ | $3.7 \times 10^{-1}$ |
| Patapsco[10] | $8.86 \times 10^{-1}$ | $3.5 \times 10^{-2}$ | $5.4 \times 10^{-4}$ | $6.09 \times 10^{-3}$ | $6.63 \times 10^{-3}$ | $7.80 \times 10^{2}$ | $5.4 \times 10^{-1}$ |
| Magothy[10] | $6.61 \times 10^{-1}$ | $2.6 \times 10^{-2}$ | $1.2 \times 10^{-4}$ | $4.54 \times 10^{-3}$ | $4.66 \times 10^{-3}$ | $4.76 \times 10^{1}$ | $4.5 \times 10^{-1}$ |
| Magothy[10] | $1.73 \times 10^{0}$ | $6.9 \times 10^{-2}$ | $1.6 \times 10^{-3}$ | $1.19 \times 10^{-2}$ | $1.35 \times 10^{-2}$ | $4.76 \times 10^{1}$ | $5.4 \times 10^{-1}$ |
| Magothy[10] | $3.69 \times 10^{-1}$ | $1.5 \times 10^{-2}$ | $1.8 \times 10^{-4}$ | $2.54 \times 10^{-3}$ | $2.72 \times 10^{-3}$ | $4.76 \times 10^{1}$ | $7.4 \times 10^{-1}$ |
| Magothy[10] | $2.40 \times 10^{-1}$ | $9.6 \times 10^{-2}$ | $3.5 \times 10^{-5}$ | $1.65 \times 10^{-3}$ | $1.68 \times 10^{-3}$ | $4.76 \times 10^{1}$ | $8.1 \times 10^{-1}$ |
| Rhode[10] | $2.49 \times 10^{-1}$ | $1.0 \times 10^{-2}$ | $2.3 \times 10^{-5}$ | $1.71 \times 10^{-3}$ | $1.74 \times 10^{-3}$ | $1.45 \times 10^{1}$ | $1.3 \times 10^{-1}$ |
| Rhode[10] | $8.80 \times 10^{-1}$ | $3.5 \times 10^{-2}$ | $3.7 \times 10^{-4}$ | $6.05 \times 10^{-3}$ | $6.42 \times 10^{-3}$ | $1.45 \times 10^{1}$ | $1.7 \times 10^{-1}$ |
| Rhode[10] | $5.46 \times 10^{-1}$ | $2.2 \times 10^{-2}$ | $6.3 \times 10^{-5}$ | $3.75 \times 10^{-3}$ | $3.82 \times 10^{-3}$ | $1.45 \times 10^{1}$ | $3.5 \times 10^{-1}$ |
| Rhode[10] | $1.24 \times 10^{-1}$ | $5.0 \times 10^{-3}$ | $2.1 \times 10^{-5}$ | $8.51 \times 10^{-4}$ | $8.72 \times 10^{-4}$ | $1.45 \times 10^{1}$ | $6.0 \times 10^{-1}$ |
| Corsica[10] | $6.17 \times 10^{-1}$ | $2.5 \times 10^{-2}$ | $1.3 \times 10^{-4}$ | $4.24 \times 10^{-3}$ | $4.37 \times 10^{-3}$ | $7.82 \times 10^{0}$ | $2.2 \times 10^{-11}$ |
| Corsica[10] | $3.64 \times 10^{-1}$ | $1.5 \times 10^{-2}$ | $7.7 \times 10^{-5}$ | $2.50 \times 10^{-3}$ | $2.58 \times 10^{-3}$ | $7.82 \times 10^{0}$ | $1.9 \times 10^{-1}$ |
| Corsica[10] | $7.14 \times 10^{-2}$ | $2.9 \times 10^{-3}$ | $1.8 \times 10^{-5}$ | $4.91 \times 10^{-4}$ | $5.09 \times 10^{-4}$ | $7.82 \times 10^{0}$ | $3.2 \times 10^{-1}$ |
| Corsica[10] | $3.69 \times 10^{-2}$ | $1.5 \times 10^{-3}$ | $2.7 \times 10^{-5}$ | $2.54 \times 10^{-4}$ | $2.80 \times 10^{-4}$ | $7.82 \times 10^{0}$ | $4.7 \times 10^{-1}$ |
| Yangtze[17] | $4.14 \times 10^{3}$ | $1.7 \times 10^{2}$ | $1.2 \times 10^{1}$ | $2.84 \times 10^{1}$ | $4.08 \times 10^{1}$ | $1.77 \times 10^{7}$ | $1.5 \times 10^{0}$ |

Micro- ($< 5$ mm) and macro- ($> 5$ mm) plastic numerical (# m$^{-3}$) and mass (g m$^{-3}$) concentrations reported by observational studies. Underlined macro-plastic numerical concentrations are derived from micro-plastic concentrations assuming a ratio of 0.04 between macro- and micro-plastic in count. This ratio is the average value from river studies reporting both size classes. As a comparison, Eriksen et al.[29] found similar macro- to micro-plastic numerical ratios for ocean particles, with values ranging from 0.01 to 0.12 (global average at 0.07). Underlined plastic mass concentrations are derived from numerical concentrations using average particle mass at sea[29]: 0.003 and 0.17 g respectively for micro- and macro-plastic. Results from a sensitivity analysis on these parameters is provided in Supplementary Table 1. Model data in the last two columns are the catchment characteristics: mismanaged plastic waste production inside the catchment ($M_{mpw}$) in t yr$^{-1}$ (refs 5,20) downstream of dams, and monthly averaged runoff ($R$) in mm d$^{-1}$ (ref. 21). Model data in the last two columns are corresponding to sampling month.

million tonnes per year, placing the greater catchment into third position.

Indonesia was also identified as a major contributor on the Asian continent, with four Javanese rivers being of particular concern. The Brantas, Solo, Serayu and Progo Rivers respectively emitting an estimated 38,900 (range 32,300–63,700), 32,500 (range 26,500–54,100), 17,100 (range 13,300–29,900) and 12,800 (range 9,800–22,900) tonnes of plastics per year. Overall, we computed a midpoint annual emission of 200,000 tonnes (14.2% of global total) from Indonesian rivers and streams, mainly coming from the Islands of Java and Sumatra. This result reflects the levels of population density, as well as waste

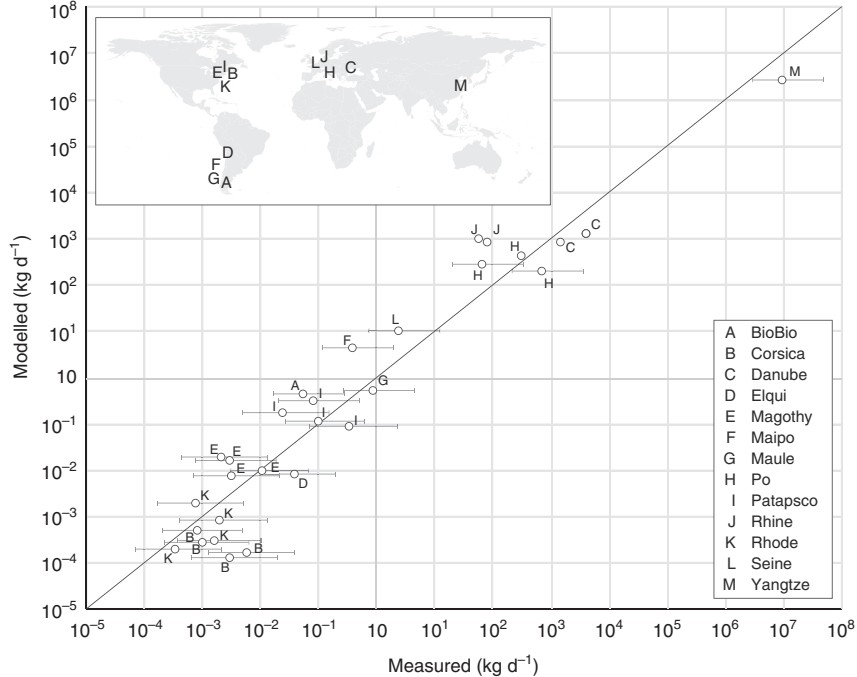

**Figure 2 | Comparison between river input model and measurements.** Circles indicate midpoint estimates of daily inputs of plastics from river to oceans in kg d$^{-1}$. Whiskers extend from lower to upper estimates based on a range of average particle mass and ratio between numerical concentrations of macro-plastics and micro-plastics. The regression analysis was carried out with 30 records from 13 rivers reported in seven studies. The locations of the 13 rivers are provided in the map.

mismanagement in the region considering the surface area of these catchments is two to three orders of magnitude smaller than other large contributing rivers in the list.

The rest of the world shared the remaining 14% of river plastic mass input, with 7.8% coming from Africa with 109,200 (range 85,700–192,000) tonnes per year, 4.8% from South America with 67,400 (range: 52,700–119,000) tonnes per year, 0.95% from Central and North America with 13,400 (range: 8,880–28,200) tonnes per year, 0.28% from Europe with 3,900 (range: 2,310–9,320) tonnes per year, and the remaining 0.02% from the Australia-Pacific region with 300 (range: 193–707) tonnes per year. In West Africa particularly, the Cross River with 40,300 (range 33,800–65,100) tonnes per year, the Imo River with 21,500 (range: 17,500–35,100) tonnes per year and the Kwa Ibo River with 11,900 (range: 9,300–20,800) tonnes per year all appeared in the list of the twenty most polluting river catchments. In South America, we estimated an annual input of 38,900 (range 32,200–63,800) tonnes per year coming from the Amazon River, the largest river on earth by water discharge, with tributaries in Peru, Columbia, Ecuador and Brazil. Also in South America, we predicted a significant contribution from the Magdalena River in Columbia with 16,700 (range 12,900–29,500) tonnes per year entering the Gulf of Mexico.

**Seasonality of river plastic inputs to oceans.** Using monthly averaged daily runoff for the period 2005–2014, we assessed seasonal variations in modelled river plastic inputs into oceans worldwide. We estimated that >74.5% of the total river plastic input occurs between May and October. Our model showed a peak in global inputs for the month of August with an estimated 228,800 (range 193,000–375,000) tonnes, and a low for the month of January with 46,200 (range 34,200–87,500) tonnes. These seasonal findings were mainly driven by the large inputs from China which are regulated by the East Asian monsoon. In the

Asian region, opposing seasons between northern and southern hemispheres create distinct monsoon regimes separated by the South China Sea[26], with the Indian and East Asian summer monsoons (June to September) in the North and the South East Asia Summer monsoon (November to March) in the south[27] (Fig. 3a).

The changes in rainfall rates associated with monsoons was reflected in the predicted monthly river plastic inputs into oceans. The Yangtze River contributed a midpoint estimate of 76,000 tonnes per month in July but <2,500 tonnes per month in January. Similarly, our model showed midpoint input estimates from the Ganges River peaking in August with 44,500 tonnes per month while the river discharges <150 tonnes per month between December and March. In Southeast Asia however, maximum river plastic inputs occurred early in the year, during the regional rainy season. In Indonesia for example, modelled inputs of the archipelago reached a midpoint estimate peak in February with 35,000 tonnes per month, as opposed to 1,800 tonnes per month in August during the dry season.

Looking at seasonal inputs worldwide, it appears that the relative variations in monthly inputs from the Asian subcontinent are not as pronounced as for other continents (Fig. 3b). This is caused by a balance between inputs from East Asia and the Indian Subcontinent during the northern hemisphere summer and contributions from Southeast Asia during the southern hemisphere summer. For other parts of the world, our model showed two distinct river plastic input peaks: one occurring between June and October for African, North and Central American rivers, and one occurring from November to May for European, South American and Australia-Pacific rivers.

## Discussion

Here we provide a global estimate of plastic emissions from rivers into the world's oceans: between 1.15 and 2.41 million tonnes per

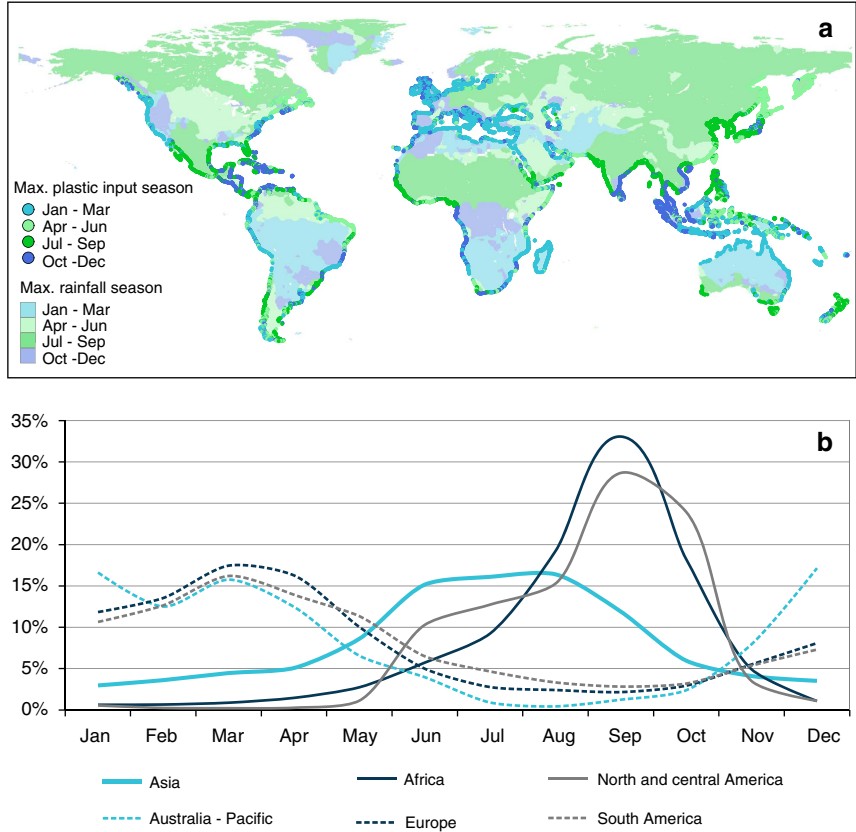

**Figure 3 | Seasonality of regional inputs of river plastic to oceans.** (**a**) River outflow locations are indicated by trimestral period when respective peak input occurs. Peak seasons for precipitation rates from GLDAS[21] are mapped on the continental landmass, showing a clear correlation with our predicted inputs. (**b**) Relative seasonality of plastic inputs from rivers into the ocean by continents. Continental contributions are expressed in percentage of respective annual mass inputs.

year. Most of this river plastic input is coming from Asia, which emphasise the need to focus monitoring and mitigation efforts in Asian countries with rapid economic development and poor waste management. Research on freshwater plastic pollution is a relatively new field and most efforts have been carried out in industrialized countries of Europe and North America. While many indicators suggest a dominant contribution of plastics from Asian countries[20], there is very little data to document these assumptions and thoroughly verify the validity of our model. Yet, the relatively high concentrations of ocean plastic found at the surface of the North Pacific Ocean[28,29] where buoyant plastics originating from Asia can accumulate[30], suggest that our assumptions are plausible.

It has been estimated that between 4.8 and 12.7 million tonnes of plastic enters the ocean every year from coastal populations worldwide[5]. Plastics within coastal areas however do not only enter the oceans through rivers. It can also reach oceans by other processes such as direct littering near beaches, followed by tidal or wind transport[4]. It is also important to note that our model is calibrated against buoyant plastics found on river surface waters, whereas this previous coastal contribution assessment considers all types of plastics found in municipal waste. Finally, we only consider a section of the full plastic debris size spectrum as particles smaller than the mesh size of the sampling nets were not accounted for and debris larger than the aperture size of the trawl devices are under-represented. For these reasons our estimate should be considered conservative. Nonetheless, for comparison purposes, we applied our model to calculate the contribution of coastal populations (that is, living within 50 km from the

coastline) and obtained a global contribution from coastal population of 356,000–893,000 tonnes per year. This suggests that at least 2.8–18.6% of the coastal plastic emissions occur via river transport. Furthermore, our findings indicate that an additional plastic input of at least 0.79–1.52 million tonnes per year reaches oceans from inland areas via rivers.

It should be stressed that there is a great level of uncertainty related to both measurements and models of plastic transport in riverine systems. Plastic sources, deposition and degradation processes are poorly understood, and may play an important role in the size and spatial distributions of plastic in freshwater ecosystems[31]. For instance, when taking into account the uncertainties related to the extrapolations we made in the field observation dataset, our global input prediction doubled from lower to upper estimates (Supplementary Table 1). Furthermore, a major assumption of our model is that artificial barriers such as dams act as sinks for all plastic particles. This is a conservative assumption since sediment trap efficiency of reservoirs may vary upon several factors[32].

There is a need for more extensive monitoring of freshwater contamination by plastic pollution with systematic samplings in sediment and surface waters of rivers throughout the year to better understand seasonality of inputs. Methodologies and reported units should be better standardized, as these vary widely across assessments. Furthermore, plastics should not be regarded as a generic material but must be classified into classes (for example, polymer types, debris sizes). Defining better standards for monitoring plastic pollution in freshwater ecosystems would allow more robust comparisons across catchments

and higher levels of sophistication of numerical models at global scale. Long-term monitoring would improve the definition of the sampled debris size spectrum. Studies should aim at providing mass estimates and not only focus on reporting numbers of particles per units of volume (or surface area). Reporting weight is critical to assess conservative budget assessment and compare estimates against global plastic production statistics. Finally, depending on precipitation patterns, soil characteristics[25] and river hydrodynamics[31], micro- and macro-plastic retention, as well as vertical distribution may vary significantly along rivers. Similarly, to open ocean waters where empirical models using sea surface wind velocities, sea state and particles properties have been proposed[33–35], an approach predicting plastic loads in the water column of rivers as a function of its physical and geological characteristics would be beneficial to account for suspended particles and refine our estimates.

Here we provide a geographical and seasonal distribution of plastic inputs from rivers into the ocean at global scale. Both our methodology and model predictions can be refined once more, data are made available. While population density and waste management data demonstrate a good correlation with results from observational river studies, freshwater plastic contamination can occur from different sources. For example, micro-plastics retained in waste water treatment plants can be reintroduced in the environment with application of sewage sludge on agricultural lands[36]. As such, our model could include multiple source scenarios such as land use characteristics and economic activities to account for the inputs from the industrial and agricultural sectors, land transportation, construction, tourism, fisheries and aquaculture industries. The modelling of local sinks for plastics on rivers could also be refined with the integration of river morphologies and local hydrodynamics (for example, sedimentation, remobilization), as well as the presence of natural (for example, mangrove) and artificial (for example, river groynes or waste water treatment plants) features. Our results are made available in Figshare[37] in an attempt to assist researchers and policy makers with selecting locations for future monitoring and mitigation efforts.

## Methods

**Model approach.** Our global model computes plastic load inputs from 40,760 watersheds worldwide[21] into the ocean using geospatial data on population density[18,19] and MPW production per inhabitant and per day in 182 individual countries[5,20]. Waste is considered mismanaged when it is littered or inadequately disposed. MPW corresponds to the fraction of plastic found in mismanaged waste material on land. This definition was applied in a previous estimate of global plastic waste inputs into the ocean from coastal population worldwide[5]. MPW production rates are integrated inside catchments. The resulting mass is accumulated following natural drainage patterns derived from local topography, and taking into account the presence of artificial barriers (for example, dams and weirs) acting as sinks. The seasonality of inputs at the outfall location is derived from monthly average catchment runoff. An empirical relation using integrated MPW mass production upstream of river mouths and seasonal runoff is formulated and calibrated using a set of field observations (Fig. 4).

**Model formulation.** To estimate daily plastic mass input from individual watersheds, we used the following parametric equation:

$$M_{out} = (kM_{mpw}R)^a,\qquad(1)$$

where $M_{out}$ is the plastic mass release at the outflow in kilogram per day, $M_{mpw}$, the mass of MPW produced inside the catchment downstream of artificial barriers and $R$, monthly averaged catchment runoff. $k$ and $a$ are the regression parameters. We find a strong coefficient of determination ($r^2 = 0.93$) for $k = 1.85\ 10^{-3}$ and $a = 1.52$ (midpoint estimate, Fig. 2) using $n = 30$ records from 13 different rivers, where data on plastic contamination in surface waters were reported in the literature.

We considered only peer-reviewed studies that provided reliable estimates of plastic concentrations (number and/or mass of plastic particles per volume and/or area of river water) using surface net devices. For plastic concentrations reported in number of particles per unit area of river surface, we used the depth of the trawling devices to convert reported surface areas ($km^2$) into volume of water sampled ($m^3$). The bibliographic review and selection criteria described above led

to the consideration of seven studies in our model calibration exercise. These studies reported river plastic concentrations in 13 rivers, at 30 sampling events that occurred in different time periods (Table 2). Our approach is conservative because we neglected the contribution of buoyant plastic that may occur below the sampled depth due to water turbulence[13]. Furthermore, this approach does not account for the contribution of non-buoyant plastics that, once introduced in rivers, may slowly make its way to the oceans due to turbulent transport, accumulating in deep sea river canyons[38]. Around 48% of the plastic produced yearly is made of polymers lighter than seawater (Polyethylene and Polypropylene;[1]), this number is likely higher due to existence of objects made of polymers heavier than seawater that can float due to air entrapment (for example, PET bottles and foamed Polystyrene).

Not all studies considered here reported micro- and macro-plastic concentrations at surface waters of rivers. As such, for our midpoint estimate, we homogenized our data set using the mean ratio micro- to macro-plastic numerical concentration from studies reporting both types (mean ratio equals to 0.04). For comparison, a study compiling thousands of samples at sea found a relatively similar mean ratio of 0.07 (ref. 29). When only numerical concentrations were reported, we estimated the mass concentrations using the average mass of micro- and macro-plastic particles sampled at sea: 0.003 and 0.17 g, respectively[29]. The results of the standardization exercise are presented in Table 3.

We acknowledge however, that the extrapolations described above are a limitation of the calibration exercise presented here, as the average mass of river plastic particles, as well as the ratio between micro- and macro-plastic concentrations may vary across catchments due to local differences in *in-situ* fragmentation rates, plastic transport processes and levels of primary micro-plastic emissions (for example, pre-production pellets, microbeads from cosmetics and hygienic products, laundry powders, paint and coating flakes). A sensitivity analysis was conducted by varying the mean ratio micro- to macro-plastic numerical concentration (range: 0.01–0.12) and the average mass of particles (range: 0.002–0.004 g and 0.04–0.33 g for micro- and macro-plastic, respectively) using range values found at sea[29]. We determined an upper and lower input estimate using equation (1) with respectively $k = 1.07\ 10^{-3}$, $a = 1.61$ ($r^2 = 0.93$, $n = 30$) and $k = 4.46\ 10^{-3}$, $a = 1.42$ ($r^2 = 0.91$, $n = 30$). Further details on the sensitivity analysis are provided in Supplementary Table 1.

**Correction for surface waters.** Some studies[8,9] directly provide an estimate of daily or yearly plastic mass input rate. For the other studies, we computed the daily releases of plastic from rivers into the ocean by multiplying the estimated mass concentrations from observations by the volume of water flowing at the surface layer per day. For each river, the surface layer thickness was taken at the sampling depth reported by the study, therefore the contribution of any particles suspended below the sampled depth was neglected. We derived the surface layer flow from the river depth and the total monthly averaged discharge predicted by our hydrological model using the month corresponding to the surveyed time period. When the river depth was not reported by the study, we used the following relationship in equation (2) between channel form and discharge[39]:

$$D = cQ^f,\qquad(2)$$

where $Q$ is the river discharge, $D$ is the river depth, $c$ and $f$ are parameters. A good coefficient of determination ($r^2 = 0.75$) was found for $c = 0.349$ and $f = 0.341$, when comparing discharge and bed form of 674 rivers in Canada and USA[40]. When studies reported surveys directly from estuaries, the depth was estimated using nautical charts.

**Estimating MPW mass in catchments worldwide.** We combined data on waste generation in kilograms per inhabitant and per day for 182 individual countries[5,20] with gridded population densities in inhabitants per $km^2$ (refs 18,19) to estimate inland MPW production rates per year. An exception was made for Sri Lanka, where we replaced the World Bank statistics with values reported in more detailed regional assessments[41,42]. We computed a global $\frac{1}{4}$ degree resolution grid of estimated mass of MPW generation on land in tonnes per year. In this model, we assumed that inland plastic is accumulated by following natural drainage patterns, derived from the space borne elevation data[22]. The global landmass surface area was divided in river catchments from the U.S. Geological Survey Agency that are used by the Global Land Data Assimilation System (GLDAS, ref. 21). We used the flow accumulation toolset from ESRI's ArcGIS software to compute the total mass of inland MPW upstream of the outflow location. The outflow is the most downstream position in a river catchment and determines the input source point into the ocean. Input from catchments with an outflow not connected to the ocean (for example, specifically arid inland areas) were discarded. The model takes into account the presence of artificial barriers and treats them as accumulation sinks, where plastic at the surface is intercepted. As a result, the predicted plastic concentration at the river mouth is representative of the accumulation of inland MPW (in tonnes per year) in the catchment area downstream of artificial dams.

The consideration of dams in our numerical model was motivated by a better correlation found with measurements (Table 4) than when including MPW production rates upstream of dams. Artificial barriers in rivers may retain 65% of the global input into freshwater, as we calculated an annual 2.13–4.46 million tonnes of plastic introduced upstream of dams that are not accounted as input into

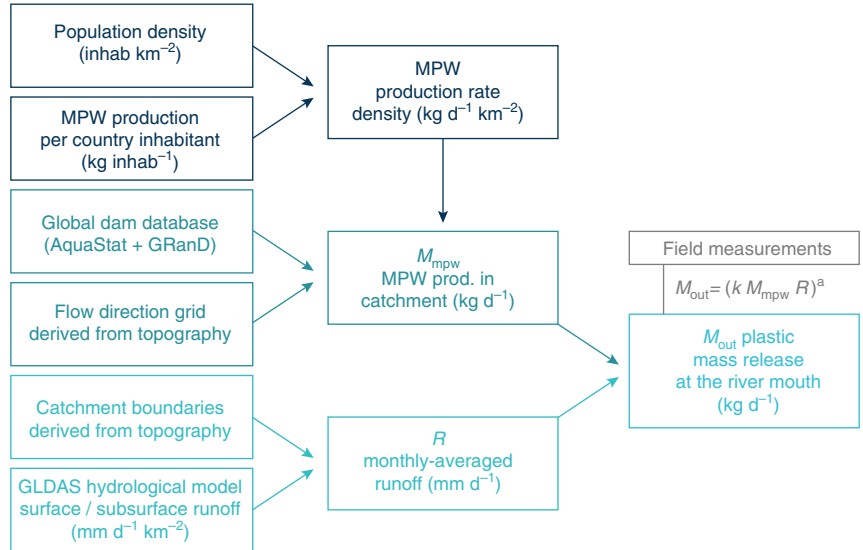

**Figure 4 | Model framework.** Plastic mass production per river catchment ($M_{mpw}$; $n = 40,760$ rivers) is computed from data on MPW production rates per country, population density, topographic elevation and location of artificial barriers. Seasonality of inputs is derived from monthly averaged runoff ($R$). A parametric equation with parameters $k$ and $a$ is used to fit model predictions ($M_{out}$) against results from observational studies. For our mid-point estimate, best fit was found for $k = 1.85 \times 10^{-3}$ and $a = 1.52$ ($r^2 = 0.93$, $n = 30$).

**Table 4 | Pearson's product moment correlations of total mass concentration measurements with watershed characteristics.**

| Catchment characteristics | With outliers (n = 30) | | Without outliers (n = 29) | |
|---|---|---|---|---|
| | r | p | r | p |
| $M_{mpw}$ | 0.9985 | **<0.001** | 0.1934 | 0.314 |
| $M_{mpw}$ downstream of dams | 0.9988 | **<0.001** | 0.3555 | 0.058 |
| $R_{year}$ | 0.0209 | 0.9128 | 0.1713 | 0.374 |
| $R_{month}$ | 0.4268 | **0.019** | 0.3128 | 0.099 |
| $M_{mpw}$ x $R_{month}$ | 0.9999 | **<0.001** | 0.1741 | 0.366 |
| $M_{mpw}$ downstream of dams x $R_{month}$ | 0.9999 | **<0.001** | 0.4132 | **0.026** |

Mismanaged plastic waste production inside a catchment ($M_{mpw}$) is expressed in $t\,yr^{-1}$ (with and without the consideration of dams), monthly averaged and yearly averaged catchment runoff ($R_{month}$ and $R_{year}$) in mm d$^{-1}$. Correlation coefficients ($r$) are positive for all characteristics. Pairs with P values below 0.05 are statistically significant (shown in bold). Weight concentrations ($n = 30$) demonstrated an excellent correlation with $M_{mpw}$ with $r$ coefficient near 1 and P values near 0, however this result may be biased by the inclusion of measurements from the Yangtze River which may be seen as an outlier with at least three orders of magnitudes difference with the rest of reported concentrations. When removing outliers, the best correlation is found with the product of monthly averaged catchment runoff $R_{month}$ and $M_{mpw}$ downstream of dams (statistically significant positive correlation, $n = 29$, $r = 0.4132$ and $P = 0.026 < 0.05$).

the oceans by our model. These results were calculated using the parametric equation determined when considering MPW downstream of dams as model proxy. Including MPW production upstream of dams, when assessing the linear regression would result in different model parameters $k$ and $a$ in equation (1). While determining regression coefficients based on MPW quantities upstream of dams, our model predicted a global input of 0.76–1.55 million tonnes per year (midpoint at 0.91 million tonnes per year) which remains in the same order of magnitude than the initial scenario. The decrease in predicted global input from the current model may be explained by the number of dams present in the large rivers covered by the observational studies. In the Yangtze River and Danube River catchments particularly, respectively 68% and 78% of MPW production occurs upstream of a dam. Therefore, relative MPW mass have less weight on the overall prediction result which ultimately leads in a decrease of our global estimate.

Dam locations were derived from the United Nation Food and Agriculture Organization's AquaStat dam database[23], consisting of 8,800 dams worldwide with a minimum height of 15 m or a reservoir capacity of >3 million m$^3$. The Global Rivers and Dam Database (GRanD database; ref. 24) was used for South America as the AquaStat database was incomplete for this continent. The catchments containing the largest number of dams were the basins of the Mississippi River (718 dams), Yangtze River (342 dams) and Danube River (184 dams). As the analysis is based on natural drainage patterns, the model limitations are that man-made channels are not taken into account and that plastic load accumulates in the main arms of rivers at deltas, introducing uncertainties at local scales. These limitations however do not affect the global inputs estimate. An example for the island of Java in Indonesia illustrating the different datasets involved in this framework is provided in Fig. 5.

**Estimating monthly averaged catchment runoff.** In our model, surface runoff is included as a model parameter to account for (1) the introduction of MPW in riverine system during episodes of heavy rainfall[10] and (2) the remobilization of

deposited plastic particles during flood events[31]. Monthly averaged catchment runoff in millimetres per day was calculated using GLDAS driving the NOAH Land Surface Model[21]. This land surface modelling system integrates data from advanced ground and space-based observation systems. The model contains land surface parameters for vegetation, soil, elevation and slope. The forcing data in the model are near-real-time satellite-derived precipitation and evaporation data (wind, radiation, temperature, humidity and surface pressure). The model computes the daily surface and subsurface runoff globally on $\frac{1}{4}$ degree resolution, by solving terrestrial water and energy budgets[21]. Subsurface runoff consists of water that infiltrates into the soil and flows to a water body by groundwater flow. Surface runoff occurs either when the rainfall exceeds the infiltration capacity of the soil or when the soil is saturated with groundwater. Monthly and yearly averages are calculated over the period 2005–2015. The surface and subsurface runoff are summed and subsequently averaged per catchment area[22]. A better correlation was found with estimated flux inputs from observational studies when considering monthly averaged runoff instead of the yearly average (Table 4). Therefore, monthly averaged catchment runoff corresponding to sampling event month was considered while calibrating our model to account for temporal variations and seasonality of inputs.

The main motivation behind using runoff data from GLDAS is the provision of land surface processes including runoff estimates at a global level. Nonetheless, it is important to notice that comparisons between river discharge predictions from GLDAS and observations in 66 basins worldwide[43] demonstrated that predictions are somewhat dryer than observations. The authors of this validation study attributed the differences to uncertainties in precipitation rates. As our framework relies on intra-annual variability, the NOAH land surface model predictions, forced with GLDAS, were still in good agreement with seasonal variations measurements with a predicted date of maximum discharge within 20 days of observed annual discharge peak date for most rivers covered in the GLDAS validation study.

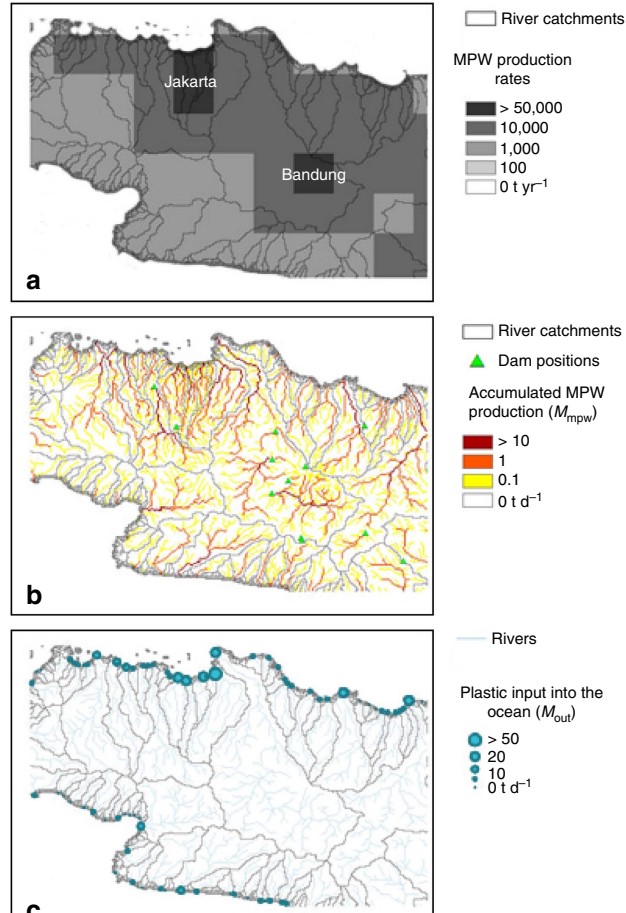

**Figure 5 | Modelled data flow illustration for Western Java, Indonesia.**
(**a**) Estimated MPW production rates in t yr$^{-1}$. (**b**) Accumulated MPW production in rivers and location of artificial barriers. (**c**) Predicted plastic mass input into the ocean at river mouths in t d$^{-1}$.

**Data availability.** The authors declare that the main data supporting the findings of this study are available within the article and its Supplementary Information. Global model inputs and outputs for lower, midpoint and upper estimates and for the 40,760 catchments considered in this study have been deposited in geospatial vector data format for geographic information system (GIS) software on figshare with the identifier doi:10.6084/m9.figshare.4725541.

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

## Acknowledgements

We would like to thank The Ocean Cleanup Foundation supporters who made the realization of this study possible. We would also like to thank three anonymous reviewers for their valuable inputs and comments on this manuscript.

## Author contributions

L.C.M.L, J.v.d.Z., J.R. and B.S. designed the study, L.C.M.L. and A.A. conducted the bibliographic review, L.C.M.L, J.v.d.Z. and J.W.D. developed the model, L.C.M.L, J.v.d.Z. and J.R. wrote the manuscript, L.C.M.L and J.v.d.Z prepared the figures. All authors reviewed the manuscript.

## Additional information

**Competing interests:** The authors declare no competing financial interests.

