## [Peer Review File · Nature Communications]

Reviewers' comments:

Reviewer #1 (Remarks to the Author):

This is an extensive research challenge the authors have attempted to tackle. It is important work to be undertaken, as the authors stated well in the introduction. And it is the first global estimate of riverine input of plastic into the world's oceans. While the global estimate "seems reasonable" (and very conservative as the authors' state – how do you know if conservative is too conservative?), the authors' also acknowledge that the uncertainty is high. I think that there should be some quantitative value on either the uncertainty or, at a minimum, a sensitivity analysis conducted on important variable so that if things were to change, we have some idea of how the results would be impacted. What are the driving variables, for example? If the authors know from doing this analysis, this should be shared. There are more details on these comments below and in the PDF as comments.

1. It is not clear in this paper that macroplastic is the only plastic being modeled (if this is a correct statement?). As pointed out in the beginning, the previous riverine studies have been a combination of macro and microplastic results. Many of them referenced were microplastic. If the only input source for this paper is mismanaged plastic waste (MPW), then comparison to field studies on microplastic do not seem appropriate for validation. This is even acknowledging the fact that secondary microplastic s are produced from the macroplastics – but the time frame and mechanisms are not well-known for this formation. It looks like the ratio of 0.04 was used to relate micro v macro – but this comes from only 2 studies?

a. Here is what the authors say: "We acknowledge however, that the extrapolations described above are a limitation of the calibration exercise presented here, as the ratio between micro- and macro-plastic concentrations may vary across catchments due to local differences in in-situ fragmentation rates, plastic transport processes, and levels of primary micro-plastic emissions (e.g. pre-production pellets, microbeads from cosmetics and hygienic products, laundry powders, paint and coating flakes)." So I understand you acknowledge the limitation of using this data, but see point 3, below, maybe a sensitivity analysis will tell us how much of an impact this assumption has.

2. If plastic is floating on the surface of a waterway, flow-over dams and spillways may not be catchment points. Were the dams characterized at all in this manner? How much of an impact are the dams on the results? (another sensitivity analysis might be warranted here).

3. Overall, a sensitivity analysis is needed. What if the 0.04 was changed – how does it impact the results? What component(s) are the driving factors – population? MPW? Dams? We need to know if various influential variables were changed, how would this impact the results? What does this mean to the results and potential uncertainties surrounding them? It should be discussed.

Reviewer #2 (Remarks to the Author):

This paper explores the spatial and temporal input of plastics to the world's oceans from freshwater sources – one of the first studies to do this. The methods seem reasonable (though with large assumptions) and the results are plausible. The results suggest that most of the plastic inputs come from Asia during the months between May and October, which could aid in mitigation. I must also note that I am a hydrologist and certainly not an expert on anything plastic.

I do have some issues with the manuscript that I think should be addressed:

[1] In the abstract, the authors do not mention where a large portion of the plastic inputs are occurring (Asia). This should be included.

[2] First sentence of the manuscript. The juxtaposition of "packing" and "fishing" is odd. I understand what the authors are trying to say, but perhaps choose examples that are more closely related.

[3] The authors discuss sources of plastics. Are plastic emissions via the air also a large source?

[4] Paragraph starting with line 55. I think it's a good idea to also include the concentrations here. It will help the reader get an idea of the concentration differences between regions. As a reader, I am not clear on what the concentrations should be.

[5] The work includes mostly large watersheds. The smaller coastal watersheds, where the transport is fast and there are less dams, should also be mentioned. I would imagine that these are a large, missing source as well.

[6] Line 83. Do the top 122 polluting rivers have any spatial coherence?

[7] I am interested in the temporal aspect of this study. Could a figure be made that shows the peak month of plastic input? Similar to Figure 1, but that shows the month of the year where the peak plastic inputs occur. It seems this could aid in mitigation. This should not replace Figure 3, but could provide more information. In N. America, for example, the east and west coasts are so hydrologically different that it may not make sense to lump these together.

[8] In the model, could the authors replace runoff with precipitation? This could perhaps reduce uncertainty from the hydrological model. Precipitation is measured everywhere (for the most part) and largely correlated with runoff.

[9] Line 241. Does this assume that all MPW is transported outside of the watershed in the absence of dams? Could some MPW just be buried?

[10] Similar to comment [9], is all MPW captured by weirs and dams (MPW transport = 0)? Is that a reasonable assumption?

[11] The section titled "Estimating monthly averaged catchment runoff". Because your model heavily relies on discharge data, the authors should state the accuracy of this discharge data.

[12] The figures and tables are all really well done.

Reviewer #3 (Remarks to the Author):

This paper presents the first assessment of microplastic load from rivers to the sea. The topic is highly relevant and it is gaining unprecedented attention during recent years. The approach presented in this paper is simple and clear, and the paper is well written. I think this is a great

study that could potentially represent a step forward in the field of microplastic research. Nevertheless, I have some concerns, which I am going to detail hereafter.

The model is based on the assumption that the only source of plastic within a catchment is mismanaged plastic waste, and therefore the amount of microplastics ending in the river network is proportional to population density. But what about sewage sludge application on agricultural land? This has been demonstrated to be a relevant source of microplastic pollution (e.g., Fytli and Zabaniotou, 2008, *Renew. Sustain. Energy Rev.* 12, 116–140; Rillig, 2012, *Environ. Sci. Technol.* 46, 6453–6454), and it could be driven by other land use-related variables, apart from population density. Furthermore, a very recent paper (Astrid et al., 2016, *Mar. Pollut. Bull.* doi:10.1016/j.marpolbul.2016.11.056) found little correlation between microplastic concentrations in coastal areas of Africa and population density.

Microplastic transport, analogously to sediment transport, is clearly driven by runoff and flow, and this is acknowledged by the model presented in this study, which uses monthly runoff to reproduce the seasonality of plastic load. However, I am not convinced that this representation is accurate enough. Assuming that plastic transport is governed by the same physics that drives sediment transport, there is a non-linear relationship between flow and microplastic load (see for example Crawford, C.G., 1991, *J. Hydrol.* 129, 331–348). Large amounts of microplastic are mobilised during large floods. Using monthly averages rather than instantaneous or daily values of runoff attenuates the temporal variability of the hydrological cycle and will inevitably lead to a bias in the model.

No estimation of the error is provided. This is discussed in the paper, but no quantitative estimation of the uncertainty is provided. Given the large uncertainty affecting both the observations and the model results, it is paramount to provide a range of values or a confidence interval, to avoid misinterpretation of the results.

There are already at least two microplastics modelling studies available in the literature: Nizzetto et al., 2016, *Environ. Sci. Process. Impacts* 18, 1050–1059; Besseling et al., 2017, *Environ. Pollut.* 220, 540–548. These papers present two catchment-scale microplastics models. These models are more complex than the one presented in this study, but nevertheless they present valuable insights. Given that they are the first papers ever published on microplastics modelling, I believe the present study should acknowledge and discuss them.

Other comments:

Line 80 to 83: It would be interesting to know what proportion of the total land the top 20 and top 122 catchments occupy, and what proportion of the total population live in those catchments.

Line 142: While I understand the need for a value of microplastic load into the oceans, I believe that an average figure could be misleading (see comment above about time variability). Can you provide some values of microplastic load for wet years and dry years, for example?

Lines 228–229: Please see the references provided above (Nizzetto et al, 2016, Besseling et al, 2016).

Line 247: In Figure 4, K is not defined in the caption.

Lines 262–265: This obviously leads to a slight underestimation of the total microplastic load, given that dam trap efficiency is usually less than 100%, especially during large floods (Brune, 1953, *Trans. AGU* 34, 407–418). This is not likely to alter the paper results too much, but it should probably be acknowledged.

Lines 284–298: What about the goodness of fit of the land surface model results? How reliable are they?

Replies to Referee #1 Comments

Comment 1. *This is an extensive research challenge the authors have attempted to tackle. It is important work to be undertaken, as the authors stated well in the introduction. And it is the first global estimate of riverine input of plastic into the world's oceans. While the global estimate "seems reasonable" (and very conservative as the authors' state – how do you know if conservative is too conservative?), the authors' also acknowledge that the uncertainty is high. I think that there should be some quantitative value on either the uncertainty or, at a minimum, a sensitivity analysis conducted on important variable so that if things were to change, we have some idea of how the results would be impacted. What are the driving variables, for example? If the authors know from doing this analysis, this should be shared. There are more details on these comments below and in the PDF as comments.*

Reply 1. Thank you for reviewing our manuscript. We have taken your suggested changes into account and we trust our text is now much clearer and with a more robust consideration of our model uncertainties. We have added a sensitivity analysis focussing on the model calibration parameters in Supplementary Information. We now provide global and individual estimate ranges throughout the manuscript based on the uncertainties highlighted in the sensitivity analysis. Also, we have re-ordered the subsections in Methods to clarify the model formulation, differentiate calibration parameters from model proxies and better highlight the driving variables of our framework.

Comment 2. *It is not clear in this paper that macroplastic is the only plastic being modeled (if this is a correct statement?). As pointed out in the beginning, the previous riverine studies have been a combination of macro and microplastic results. Many of them referenced were microplastic. If the only input source for this paper is mismanaged plastic waste (MPW), then comparison to field studies on microplastic do not seem appropriate for validation. This is even acknowledging the fact that secondary microplastics are produced from the macroplastics – but the time frame and mechanisms are not well-known for this formation. It looks like the ratio of 0.04 was used to relate micro v macro – but this comes from only 2 studies? a. Here is what the authors say: "We acknowledge however, that the extrapolations described above are a limitation of the calibration exercise presented here, as the ratio between micro- and macro-plastic concentrations may vary across catchments due to local differences in in-situ fragmentation rates, plastic transport processes, and levels of primary micro-plastic emissions (e.g. pre-production pellets, microbeads from cosmetics and hygienic products,*

laundry powders, paint and coating flakes).” So I understand you acknowledge the limitation of using this data, but see point 3, below, maybe a sensitivity analysis will tell us how much of an impact this assumption has.

Reply 2: Our study considers both micro- and macro-plastics. This is now clearer in the revised manuscript (see lines 91-92 and lines 284-286 of the revised text). We acknowledge however that primary micro-plastics (e.g plastic pellets, microbeads, etc) are not being taken into account in our model proxies. Such limitation is highlighted in our manuscript (lines 233-239 and lines 292-297 of the revised text). Nonetheless, we know from both field observations and plastic production statistics that the bulk of river plastic mass comes from mismanaged plastic waste (MPW). This assumption is further supported by the good correlation we found between our model outputs and measurements from river studies. Therefore, the addition of primary micro-plastic sources in our model is unlikely to change the broad global mass input patterns we report and discuss in this study.

In relation to the ratio of 0.04 used to relate micro to macro, it indeed came from two studies that reported concentrations for both types in the Danube, Rhine and Po rivers (Lechner et al. 2014, Van der Wal et al. 2015). We had to use this ratio in order to extrapolate findings of river studies focusing on different size ranges of plastic debris. Without such assumptions, we would not be able to homogenise our correlation dataset. We agree however that the assumptions made on plastic contamination characteristics while homogenising our training dataset is likely to yield significant uncertainties whether it is the ratio of concentration between micro- and macro- (the 0.04 ratio) or the average mass of particles (initially 0.03 g and 0.17 g for micro- and macro-plastics, respectively). As such, we followed your suggestion to add a sensitivity analysis to identify the impact that these assumptions have on our global input result. By varying the three parameters mentioned above (micro/macro concentration ratio, mass of micro-plastics, and mass of macro-plastic particles) in ranges found for micro- and macro-plastics at sea, we defined an upper and lower scenario to complement our midpoint estimate and introduce uncertainty ranges. We have introduced these ranges throughout the manuscript and added two additional columns for lower and upper estimate in Table 1.

Comment 3. *If plastic is floating on the surface of a waterway, flow-over dams and spillways may not be catchment points. Were the dams characterized at all in this manner? How much of an impact are the dams on the results? (another sensitivity analysis might be warranted here).*

Reply 3. While recent modelling results suggest that particles are likely to be deposited in low hydrodynamic sections of rivers such as dam reservoirs, we acknowledge that plastic trap efficiency

is not always 100%. Nonetheless, when our models consider dams as catchment points, model outputs demonstrate a better correlation with the field observations dataset (Pearson's product moment correlation test, $n=29$, $r=0.41$, $p=0.026$ without dams and $r=0.17$, $p=0.366$ with dams). We are now presenting the results of this correlation test in the manuscript (see Table 4) to better support the use of dams as one of our model proxies. Furthermore, we have (1) added a new reference to the Discussion section on the efficiency of dams (see lines 211-212 of the revised manuscript) and (2) are presenting additional results with consideration of MPW production upstream of dams (see lines 338-352 of the revised manuscript).

Comment 4. *Overall, a sensitivity analysis is needed. What if the 0.04 was changed – how does it impact the results? What component(s) are the driving factors – population? MPW? Dams? We need to know if various influential variables were changed, how would this impact the results? What does this mean to the results and potential uncertainties surrounding them? It should be discussed.*

Reply 4. Changed accordingly. We are now providing a sensitivity analysis on the calibration parameters used to determine the regression function in Equation (1). We are also defining a lower and upper scenario range to reflect uncertainties on the nature of plastic contamination (e.g. see Table 1) and providing additional results on correlation tests to justify the choice of deterministic model proxies (see Table 4).

Additional comments (from PDF provided):

Comment 5. *Line 16-19 I recommend adding references on quantities produced worldwide, and also reference the other statements with literature - if there is not a limit on the number of references.*

Reply 5. Changed accordingly. We have now included a few references in the first paragraph of the revised manuscript and are also providing information on global production quantity (> 300 million tonnes per year) in the first sentence of the manuscript.

Comment 6. *Line 26 remove thus*

Reply 6. Changed accordingly.

Comment 7. Line 79 *A general "over" is not specific enough for technical writing - if it is truly more (but rounded?), it should be reported statistically. In addition, this result should either be a range if it could be more or have an error around it. Is there any way to know the range of accuracy of this data? It is obviously not an exact number for the estimate. A range or error surrounding it should be given if it is possible.*

Reply 7. Changed accordingly. We re-worded this sentence to: "We estimated that between 1.15 and 2.41 million tonnes of plastic currently flows from the global riverine system into the oceans every year".

Comment 8. Line 111 *(kg/d) - which is really a flux input, not a mass concentration measurement.*

Reply 8. Changed accordingly. We replaced "concentration measurements" to "flux inputs derived from measurements".

Comment 9. Line 112 *As stated in summary comments, because of the differences in predicting with only macrolitter and comparisons to microplastic (primary and secondary?) -there should be a sensitivity analysis so we know the impact of the assumptions made to create this correlation (using the 0.04 ratio). e.g., how would this r-squared value be impacted if the underlying assumptions of the model changed?*

Reply 9. Changed accordingly. We are now providing r-square values for different combinations of numerical concentration ratio and particles masses in Supplementary Table 1. Furthermore, we added a paragraph in "Model Formulation" of the Method section discussing the sensitivity analysis and the impact of this assumption by introducing an upper and lower estimate range (See lines 292-302 of the revised text).

Comment 10. Line 176 *(Figure 3 caption) So this means each line is equal to 100%? Or is it each continents contribution of the total global input and then over seasons. This is more what it looks like, but the figure caption is not entirely clear on which it is.*

Reply 10. Yes, that is correct, each individual lines integrate to 100%. We added the following to the caption to make this clear: "Continental contributions are expressed in percentage of respective annual mass inputs."

Comment 11. Line 180 *remove over*

Reply 11. The first sentence of the Discussion section was changed to: *“In this study, we provide the first global estimate of plastic emissions from rivers into the world’s oceans: between 1.15 and 2.41million tonnes per year.”*

Comment 12. Line 194-199 *It seems like your contribution of riverine input within 50 km of the coastline is one component of the 15 - 40% input - or one component of the 4.8 - 12.7 - so in your case, it is 477,000 or about 10% of the lowest contribution is from floatable plastic waste going into rivers and then entering the ocean. The way this is described it sounds surprising that your estimate is less when it seems logical that it would be less (and a validation of your model) since plastic enters the ocean through direct coastal input (beaches) from littering and wind, etc. It seems like this is not taken into account in your model. But if it is (it is not clear), and then maybe your result is surprising. Based upon logic though, it should be a small portion of the Jambeck et al. number, which it is. And Beyond 50 km, your data is a new input on top of Jambeck et al. data - is that correct?*

Reply 12. Yes, this is correct. Our estimate of riverine contribution within 50 km of the coastline is indeed only one component of the total plastic input coming from coastal regions and the data beyond 50 km can be considered a new input on top of Jambeck et al. estimate. We have now re-phrased this paragraph (see lines 190-203 of the revised manuscript) to highlight the points you mention here.

Comment 13. Line 200 *So I recommend all of these caveats go first and then do the comparison, which may be more logical.*

Reply 13. Changed accordingly.

Comment 14. Line 206 *Can any of this be quantified? I think some of this needs to be quantified to be published - either through error bars on analyses or at least sensitivity analyses.*

Reply 14. Changed accordingly. Our revised manuscript is now better accounting for the uncertainties associated with the data used to calibrate our river plastic emission model. We are now providing error bars in Figure 2 and estimate ranges throughout the manuscript. Furthermore, we are better highlighting the uncertainties related to our model proxies in the Method section, including (1) MPW production rates downstream and upstream of dams and (2) predicted runoff from the land surface model (see lines 338-352 and 384-391 of the revised manuscript).

Comment 15. Line 238 *How many countries were included in this analysis?*

Reply 15. 182 countries. This information is now provided in the first sentence of the method section (see lines 249 and 321 of the revised manuscript).

Comment 16. Line 242 *So 100% of the MPW enters the river? This assumption needs to be clarified and is not likely to be the case. Some percentage of the MPW will not enter the river for various reasons - even if it is in the catchment.*

Reply 16. No, MPW production rates are accumulated following natural drainage patterns to estimate MPW mass pressure upstream of river outflows. This quantity is then used as a proxy to describe heterogeneities of freshwater contamination between catchments. We acknowledge that using the term ‘transported’ here can be misleading and so we replaced it with ‘accumulated’. We also added “using integrated MPW mass production upstream of river mouths and seasonal runoff”, as well as “Input from catchments with an outflow not connected to the ocean (e.g. specifically arid inland areas) were discarded.” to the Methods section to make this clearer (see lines 256-257 and lines 331-333 of the revised manuscript).

Comment 17. Line 255 *So the World Bank waste production data was used for every country except Sri Lanka? Did the World Bank have data for all of the countries? Then the % mismanged was taken from Jambeck et al.? Or was any of the waste data taken from Jambeck et al.? There were waste predictions made in Jambeck et al. when World Bank data was not available it seemed.*

Reply 17. When no World Bank data was available, we used the regional economic status data from Jambeck et al. 2015. We have now added a reference to Jambeck et al. 2015 alongside the original world bank dataset to this sentence.

Comment 18. Line 263 river surface?

Reply 18. Yes. Changed accordingly.

Comment 19. Line 265 *This is where it is not clear if 100% of the waste upstream of a dam is not ever getting to the ocean?*

Reply 19. In the scenario presented in the main manuscript, this is indeed our assumption. We have re-phrased this paragraph and we trust this is clearer now. We have considered dams as sinks for plastics because this scenario led to better correlations with field observations than when considering MPW production rates upstream of dams. We are now presenting all model scenarios considered alongside their correlations with field measurements to better justify our choice (see Table 4).

Comment 20. Line 275 *How do you know this? Only if you do a sensitivity analysis will you know if they impact it or not.*

Reply 20. We have deleted the word 'regional' from this sentence. The global inputs estimate will not be impacted by the relative distribution of local delta arms.

Comment 21. Line 290 *What is meant by subsurface runoff? Is it natural ground water or piped stormwater? This is not clear. And if it is recharged groundwater, it should not a flow pattern for plastics it seems like.*

Reply 21. We don't see runoff directly as a flow pattern for plastic. In this study, use this as a model proxy / estimator, as it has a good correlation with both rainfall and discharge.

In the GLDAS dataset the total runoff is distributed into surface runoff and subsurface runoff. Surface runoff occurs either when the rainfall exceeds the infiltration capacity of the soil or when the soil is saturated with water (this occurs for instance in floodplains). Urban runoff through sewage systems is not taken into account in the GLDAS model, but is included as either surface or subsurface runoff.

Comment 22. Line 294 *How was it used for calibration? Did you change this input based upon the comparison of the data to the published data?*

Reply 22. We used the date of sampling reported in individual studies and determined the corresponding monthly average runoff value. To make this clearer, we re-phrased the last sentence of this paragraph to: *"Therefore, monthly-averaged catchment runoff corresponding to sampling event month was considered while calibrating our model to account for temporal variations and seasonality of inputs."*

Comment 23. Line 306 *OK, yes, this is conservative, but is it too conservative? If only PE, PP, and foamed PS float, how much plastic are you missing? Maybe you should show how conservative this is through production statistics or something like that?*

Reply 23. Changed accordingly. We have added a sentence (lines 279-282 of the revised manuscript) stating how much plastic we are likely to be missing based on production data provided by Plastics-Europe (2016).

Comment 24. Line 344 (Table 1 caption) *Is this calculated or should this be referenced from where it came? Reference where this data came from.*

Reply 24. Changed accordingly. We added a reference to the GLDAS dataset for the two last columns of Table 1 (catchment surface area and yearly average discharge) which are not a result of our model calculations.

Comment 25. Line 349 (Table 2 caption) *This is an important table and it is not clear from Figure 2 that the data comes from only 7 studies/papers. The n=30 is for 13 different river inputs (per figure 2), but this is only 7 different studies? Maybe just make this clear in Figure 2 as well with a little note.*

Reply 25. Changed accordingly. We are now clearly stating the number of studies in Figure 2: *“The regression analysis was carried out with 30 records from 13 rivers reported in seven studies.”*

Comment 26. Line 352 (Table 2) *This is where the concern over the comparison of mismanaged waste to microplastic is evident.*

Reply 26. We trust the paucity of data is now better highlighted throughout the manuscript, figures and tables. Furthermore, we are now taking into consideration the field observations uncertainties and using them to formulate estimate ranges.

Comment 27. Line 355 (Table 3 caption) *This is good this is in the figure. Is it only two studies that have both? So that 0.04 is just from 2 studies? This question comes from looking at Table 2*

Reply 27. Yes, this is correct: 2 studies with 6 records. We are now using the 0.04 ratio value for our midpoint estimate and are providing the results of a new sensitivity study in Supplementary Table 1, where the impact of variations in this ratio (0.01 - 0.12, based on ranges found at sea) on the global input estimate is shown.

Comment 28. Line 362 (Table 3) *It should be clear where this data came directly from literature or based on literature and where it is calculated. Literature should be referenced.*

Reply 28. Changed accordingly. References next to individual river names were added in the table and values that were estimated from measurements were underlined. We are also referencing the sources of MPW and rainfall data in the legend.

Comment 29. Figure 4 (model flow) *The entire population of the country is used to produce these results? So each country is entirely divided up into the river catchments? There is no area of a country that is not covered by the catchment database? Just make sure this is clearly stated somewhere.*

Reply 29. Yes, this is correct: the entire landmass is divided into individual catchments. We added the following in the Method section (lines 327-329 of the revised manuscript): *“The global landmass surface area was divided in river catchments from the U.S. Geological Survey Agency (USGS) that are used by the Global Land Data Assimilation System (GLDAS)”*. Yet some areas on land may not directly be in a catchment connected to sea, particularly in arid regions. The contribution of populations living in these areas is not considered as oceanic input. We have added a sentence to clarify this (lines 331-333 of the revised manuscript).

Replies to Reviewer #2 Comments

Comment 1. *This paper explores the spatial and temporal input of plastics to the world's oceans from freshwater sources – one of the first studies to do this. The methods seem reasonable (though with large assumptions) and the results are plausible. The results suggest that most of the plastic inputs come from Asia during the months between May and October, which could aid in mitigation. I must also note that I am a hydrologist and certainly not an expert on anything plastic.*

I do have some issues with the manuscript that I think should be addressed:

[1] In the abstract, the authors do not mention where a large portion of the plastic inputs are occurring (Asia). This should be included.

Reply 1. Changed accordingly. We have re-phrased one of the abstract sentences to provide this information: *“The top 20 polluters, mostly located in Asia, accounted for 67% of the global total.”*

Comment 2. *[2] First sentence of the manuscript. The juxtaposition of “packing” and “fishing” is odd. I understand what the authors are trying to say, but perhaps choose examples that are more closely related.*

Reply 2. Based on this suggestion and Comment 5 of Reviewer 1, we have decided to remove these terms from the first sentence of the revised manuscript.

Comment 3. *[3] The authors discuss sources of plastics. Are plastic emissions via the air also a large source?*

Reply 3. Light plastics such as films and synthetic fibres may be deposited directly at sea via atmospheric transport (see *Dris et al. 2016 Synthetic fibers in atmospheric fallout: A source of microplastics in the environment? Marine Pollution Bulletin 104(1-2): 290-293.*). We are not aware of any general quantification of inputs into the ocean available for this process however. Nonetheless, we are now mentioning in the first paragraph that atmospheric deposition is a pathway for plastics to get into oceans.

Comment 4. *[4] Paragraph starting with line 55. I think it's a good idea to also include the concentrations here. It will help the reader get an idea of the concentration differences between regions. As a reader, I am not clear on what the concentrations should be.*

Reply 4. Changed accordingly. In this paragraph, we have now included reported concentration ranges for rivers in the Chesapeake Bay, along the Chilean coast as well as for the Yangtze River.

Comment 5. *[5] The work includes mostly large watersheds. The smaller coastal watersheds, where the transport is fast and there are less dams, should also be mentioned. I would imagine that these are a large, missing source as well.*

Reply 5. The dataset from USGS includes smaller coastal watersheds. As such, they were considered in this study. It is important to highlight however that processes such as direct oceanic inputs through littering near beaches were not taken into account. This is now better highlighted in the 2nd paragraph of the Discussion section, where we compare our river input estimations with those obtained by a previous study for coastal areas inputs (< 50 km from coastline).

Comment 6. *[6] Line 83. Do the top 122 polluting rivers have any spatial coherence?*

Reply 6. We have added a sentence to the first paragraph of the Results describing the distribution of the top 122 rivers by continent: 103 in Asia, 8 in Africa, 8 in Central/South America and 1 in Europe)

Comment 7. *[7] I am interested in the temporal aspect of this study. Could a figure be made that shows the peak month of plastic input? Similar to Figure 1, but that shows the month of the year where the peak plastic inputs occur. It seems this could aid in mitigation. This should not replace Figure 3, but could provide more information. In N. America, for example, the east and west coasts are so hydrologically different that it may not make sense to lump these together.*

Reply 7. Changed accordingly. Our Figure 3 has now 2 panels to include a figure that better depict the geography of seasons. We grouped months by trimestral period to produce a clear, easy-to-understand figure. River outflows are plotted with the trimestral period during which predicted peak input occurs. For illustration purposes, we also overlaid the landmass with similar binning but looking at precipitation rates from GLDAS, showing a good spatial correlation with our predictions.

Comment 8. *[8] In the model, could the authors replace runoff with precipitation? This could perhaps reduce uncertainty from the hydrological model. Precipitation is measured everywhere (for the most part) and largely correlated with runoff.*

Reply 8. We have decided to use the runoff as a model proxy as it may account for the amount of plastic introduced in the river system during large precipitation events and the mobilization of plastic within the river system during high discharge events.

In our model, the runoff is included as model parameter/proxy to account for 2 physical mechanisms: a) we assume that the amount of mismanaged waste that ends up in the river system is depending on the catchment precipitation. Precipitation is the dominant term in the hydrological cycle calculation and runoff is consequently highly correlated to precipitation (see Figure 3A for illustration). b) Large amounts of plastic particles may be mobilized during flood events (see additional references in the manuscript: Nizetto et al. 2016, Besseling et al. 2016). There must be a relation between river discharge, correlated to runoff and plastic load. This consideration is important when including the seasonal variability of plastic inputs.

Comment 9. [9] Line 241. *Does this assume that all MPW is transported outside of the watershed in the absence of dams? Could some MPW just be buried?*

Reply 9. No, in this framework, MPW production rates are used as a deterministic model proxy. We compute the MPW mass production at individual river outflows by looking at the landmass upstream of outflow locations and downstream of dams. We agree that the term “transport” may be misleading here and we replaced it with ‘accumulated’.

We have also adapted the last sentence of this paragraph to clarify that MPW mass production in catchment is used as a proxy in an empirical equation and is not ‘transported’ in a physical sense using mass balance equations. *“An empirical relation using integrated MPW mass production upstream of river mouths and seasonal runoff is formulated and calibrated using a set of field observations”* (see lines 256-257 of the revised manuscript).

Comment 10. [10] *Similar to comment [9], is all MPW captured by weirs and dams (MPW transport = 0)? Is that a reasonable assumption?*

Reply 10. Yes, we built our model proxy assuming that MPW transport upstream of dams and weirs is equal to zero. Our model initially did not consider dams but we decided to include them at a later stage due to a better correlation with our field observation dataset (n=30 records) when treating them as plastic sinks. We have now added results for the correlation tests for MPW production with and without considering dams in Table 4 to justify our motivation.

Comment 11. [11] *The section titled “Estimating monthly averaged catchment runoff”. Because your model heavily relies on discharge data, the authors should state the accuracy of this discharge data.*

Reply 11. Changed accordingly. We have added a paragraph discussing the uncertainty related to the GLDAS forcing database in the Method section, alongside a new reference to a validation study (lines 384-391 of the revised manuscript)

Comment 12. [12] The figures and tables are all really well done.

Reply 12. Thank you.

Replies to Reviewer #3 Comments

Comment 1. *This paper presents the first assessment of microplastic load from rivers to the sea. The topic is highly relevant and it is gaining unprecedented attention during recent years. The approach presented in this paper is simple and clear, and the paper is well written. I think this is a great study that could potentially represent a step forward in the field of microplastic research. Nevertheless, I have some concerns, which I am going to detail hereafter.*

*The model is based on the assumption that the only source of plastic within a catchment is mismanaged plastic waste, and therefore the amount of microplastics ending in the river network is proportional to population density. But what about sewage sludge application on agricultural land? This has been demonstrated to be a relevant source of microplastic pollution (e.g., Fytali and Zabaniotou, 2008, *Renew. Sustain. Energy Rev.* 12, 116–140; Rillig, 2012. *Environ. Sci. Technol.* 46, 6453–6454), and it could be driven by other land use-related variables, apart from population density. Furthermore, a very recent paper (Astrid et al., 2016. *Mar. Pollut. Bull.* doi:10.1016/j.marpolbul.2016.11.056) found little correlation between microplastic concentrations in coastal areas of Africa and population density.*

Reply 1. In this framework, we find that mismanaged plastic waste (MPW) production rates inside catchment is a good descriptor for the heterogeneities of material input observed in rivers globally. We believe this occurs due to the far higher plastic mass coming from MPW when compared to other sources such as sewage sludge. Nonetheless, we are now better acknowledging that plastic within river catchments is not only coming from mismanaged plastic waste production (lines 233-236 of the revised manuscript) by adding a specific example on the topic of sewage sludge (citing Zubris and Richards, 2005). In this same paragraph, we also state that as more data is made available, we can challenge our results and increase the level of sophistication of our model by integrating new sources. In relation to the Astrid et al. 2016 study, the authors report contamination levels along coastline sediments and surf zone waters (not in river mouths). The lack of correlation with population density was attributed to the action of oceanic currents redistributing the plastics in coastal environments.

Comment 2. *Microplastic transport, analogously to sediment transport, is clearly driven by runoff and flow, and this is acknowledged by the model presented in this study, which uses monthly runoff to reproduce the seasonality of plastic load. However, I am not convinced that this representation is*

accurate enough. Assuming that plastic transport is governed by the same physics that drives sediment transport, there is a non-linear relationship between flow and microplastic load (see for example Crawford, C.G., 1991. J. Hydrol. 129, 331–348). Large amounts of microplastic are mobilised during large floods. Using monthly averages rather than instantaneous or daily values of runoff attenuates the temporal variability of the hydrological cycle and will inevitably lead to a bias in the model.

Reply 2. We agree that using monthly average runoff values may lead to some bias on our results. However, knowing the uncertainty related to estimating daily plastic mass flux from measurements (see sensitivity analysis in Supplementary Table 1), we considered that our field observations dataset is too small (n=30) to satisfactorily reproduce daily events. With such a lack of ground-truth data, going down to this level of temporal resolution would inevitably increase the level of uncertainties of our results. Nevertheless, we are now acknowledging recent modelling efforts on the effects of hydrodynamics during flood events and remobilization of deposited micro- and macro-plastic particles (additional references: Nizzetto et al. 2016 and Besseling et al. 2016; see lines 98, 208, 224-226 and 368 of the revised manuscript).

Comment 3. *No estimation of the error is provided. This is discussed in the paper, but no quantitative estimation of the uncertainty is provided. Given the large uncertainty affecting both the observations and the model results, it is paramount to provide a range of values or a confidence interval, to avoid misinterpretation of the results.*

Reply 3. We are now providing lower and upper ranges to our initial midpoint estimates. These reflect uncertainties related to the nature of plastic contamination and the calibration parameters used to homogenise the observational studies dataset. Results from a sensitivity analysis are provided in Supplementary Table 1. Range values were systematically included throughout the revised manuscript and in Table 1.

Comment 4. *There are already at least two microplastics modelling studies available in the literature: Nizzetto et al., 2016. Environ. Sci. Process. Impacts 18, 1050–1059; Besseling et al., 2017. Environ. Pollut. 220, 540–548. These papers present two catchment-scale microplastics models. These models are more complex than the one presented in this study, but nevertheless they present valuable insights. Given that they are the first papers ever published on microplastics modelling, I believe the present study should acknowledge and discuss them.*

Reply 4. Changed accordingly. We are now acknowledging both studies by referencing them multiple times in the Discussion and Method sections (see lines 98, 208, 224-226 and 368 of the revised manuscript).

Comment 5. *Line 80 to 83: It would be interesting to know what proportion of the total land the top 20 and top 122 catchments occupy, and what proportion of the total population live in those catchments.*

Reply 5. Changed accordingly. This information is now provided in the first paragraph of the Results section (see lines 83-89 of the revised manuscript).

Comment 6. *Line 142: While I understand the need for a value of microplastic load into the oceans, I believe that an average figure could be misleading (see comment above about time variability). Can you provide some values of microplastic load for wet years and dry years, for example?*

Reply 6. Changed accordingly. We have added ranges for lower and upper estimates throughout the revised manuscript for global input and individual river contributions to reflect the uncertainties related to our method. We are also providing a new map (see Figure 3b) where the seasonal patterns on river plastic emissions to the world's ocean are better visualised. In regards to estimating contributions for wet and dry years, we believe our field observations dataset is too small to reasonably assess inter-annual variations. Systematic monitoring in rivers throughout the year, for several years would assist in answering these questions and help us refine our model assumptions in the future.

Comment 7. *Lines 228-229: Please see the references provided above (Nizzetto et al, 2016, Besseling et al, 2016).*

Reply 7. Both references were included in the revised manuscript. Furthermore, we included “*and local hydrodynamics (e.g. sedimentation, remobilization) and as well as*” to this Discussion sentence (see lines 239-244 of the revised manuscript).

Comment 8. *Line 247: In Figure 4, K is not defined in the caption.*

Reply 8. Changed accordingly. We are now defining K in the caption of Figure 4. This is one of the parameters of our parametric equation. We added the two following sentences in the caption: “A

parametric equation with parameters k and a is used to fit model predictions (M_{out}) against results from observational studies. For our mid-point estimate, best fit was found for $k = 1.85 \cdot 10^{-3}$ and $a = 1.52$ ($r^2 = 0.93$, $n = 30$)."

Comment 9. *Lines 262-265: This obviously leads to a slight underestimation of the total microplastic load, given that dam trap efficiency is usually less than 100%, especially during large floods (Brune, 1953. Trans. AGU 34, 407–418). This is not likely to alter the paper results too much, but it should probably be acknowledged.*

Reply 9. Changed accordingly. We are now acknowledging that sediment traps in dams may not always be 100% efficient and included a reference to Brune (1953) in the Discussion section (see lines 211-212 of the revised manuscript). Furthermore, we added a paragraph on the assumption of dams acting as sinks and presented correlation results when considering MPW production rates upstream of dams (see Table 4).

Comment 10. *Lines 284-298: What about the goodness of fit of the land surface model results? How reliable are they?*

Reply 10. We added a paragraph in the Method section discussing the accuracy of the GLDAS forcing dataset as well as a reference to a validation study (lines 384-391 of the revised manuscript)

REVIEWERS' COMMENTS:

Reviewer #1 (Remarks to the Author):

It appears that the initial comments have been addressed adequately.

One last comment on Figure 3a - the map of the seasonality of inputs... red is a color attributed to "more", especially compared to blues, etc. I think this figure will be misinterpreted as red being more. If it is just seasonality, I suggest green and blue - green for the summer and blue for the winter (shaded for spring and fall as you have it) - otherwise this looks like a "heat map" with higher concentrations being red. As it is, it will be glanced at and misinterpreted.

I also don't see the entire list of rivers given as a part of this - maybe I missed that, but the entire list should be given. Both for data transparency and for use by others. As a spreadsheet would be best (like Jambeck et al. have placed online), but at least in a document, if not the spreadsheet. I realize it is a long list (>47,000 entries).

Reviewer #2 (Remarks to the Author):

I am satisfied with the authors' response to my concerns as well as the other reviewers' concerns. I am especially satisfied with the additions of Figure 3 and the added text that added clarification. I think the manuscript is greatly improved.

Reviewer #3 (Remarks to the Author):

I would like to thank the authors for the effort they made to reply to my comments. I am satisfied with the current status of the paper and I recommend it is accepted for publication.

Replies to Referee #1 Comments

Comment 1. *One last comment on Figure 3a - the map of the seasonality of inputs... red is a color attributed to "more", especially compared to blues, etc. I think this figure will be misinterpreted as red being more. If it is just seasonality, I suggest green and blue - green for the summer and blue for the winter (shaded for spring and fall as you have it) - otherwise this looks like a "heat map" with higher concentrations being red. As it is, it will be glanced at and misinterpreted.*

Reply 1. The colours in Figure 3a were changed accordingly with tones of blue and green.

Comment 1. *I also don't see the entire list of rivers given as a part of this - maybe I missed that, but the entire list should be given. Both for data transparency and for use by others. As a spreadsheet would be best (like Jambeck et al. have placed online), but at least in a document, if not the spreadsheet. I realize it is a long list (>47,000 entries).*

Reply 2. The full river dataset containing 40,760 river mouth points with lower, mid and upper estimates (monthly and yearly) as well as runoff, MPW and area was deposited on figshare and will be granted open access. A reference to the dataset DOI was included in the manuscript (line 243 of the revised manuscript) as well as in the data availability statement.